# Bridging the "know-do" gap to improve active case finding for tuberculosis in India: A qualitative exploration into national tuberculosis elimination program staffs' perspectives

Hemant Deepak Shewade[1‡]*, Prabhadevi Ravichandran[1‡], S. Kiran Pradeep[1], G. Kiruthika[1], Devika Shanmugasundaram[1], Joshua Chadwick[1], Swati Iyer[2], Aniket Chowdhury[2], Dheeraj Tumu[2], Amar N. Shah[3], Bhavin Vadera[3], Venkatesh Roddawar[4], Sanjay K. Mattoo[5], Kiran Rade[2‡], Raghuram Rao[5‡], Manoj V. Murhekar[1‡]

1 ICMR – National Institute of Epidemiology (ICMR-NIE), Chennai, India, 2 Office of the World Health Organization (WHO) Representative to India, WHO Country Office, New Delhi, India, 3 USAID India, New Delhi, India, 4 John Snow India Private Ltd., New Delhi, India, 5 Central TB Division, Ministry of Health and Family Welfare, New Delhi, India

‡ HDS and PR contributed equally as primary authors. KR, RR and MVM contributed equally as senior authors.
* hemantjipmer@gmail.com, hemant@nie.gov.in

**Data Availability Statement:** Data cannot be shared publicly. In this qualitative study, the

## Abstract

### Background

In 2022, India's national tuberculosis (TB) elimination program (NTEP) commissioned a national level evaluation of active case finding (ACF) for TB to guide evidence-based strategic planning. As part of this evaluation, based on secondary data analysis we observed that the quality of ACF was suboptimal in 2021. Hence, this study aimed to understand the enablers, barriers, and suggested solutions to improve ACF for TB in India from NTEP staff (provider) perspective.

### Methods

This was a descriptive qualitative study involving key informant interviews from six districts and eight states, conducted between February and August 2023. We purposively selected key state- district- and sub-district-level program managers and implementers who were experienced and vocal. The interviews were audio recorded and transcribed verbatim by research interns and investigators. Two investigators independently did manual descriptive thematic analysis, and a third investigator resolved inconsistencies. The themes and categories emerged by collating together the results of the coding process.

### Results

A total of 34 key informant interviews were conducted and of these, four were repeat interviews. Adequate budgets for ACF including incentives, performance review mechanism,

transcripts are the data. The transcripts, even if anonymized, risk the confidentiality of the person who provided the information / perspective. In the patient information sheet and consent form that was signed, we do not have the consent to share the transcript from the interviews. Hence, sharing the transcript will breach the compliance with the protocol approved by the ICMR-NIE's institute human ethics committee. Data are available from the ICMR-NIE's institute human ethics committee (contact ihec.nie@gmail.com) for researchers who meet the criteria for access to confidential data. The anonymized transcripts are available on request to the corresponding author subject to signature of a data confidentiality agreement.

**Funding:** ICMR-NIE led USAID/JSI supported tuberculosis active case finding (TB ACF) evaluation project is a collaborative effort involving ICMR-National Institute of Epidemiology (ICMR-NIE), Chennai, India (lead); USAID India, New Delhi, India; John Snow India Private Limited, New Delhi, India; The WHO Country Office for India, New Delhi, India; and Central TB Division, Ministry of Health and Family Welfare, Government of India, New Delhi, India. This study was commissioned by India's national TB elimination program, with funding support from United States Agency for International Development (USAID) through John Snow International (JSI) under the TB Implementation Framework Agreement (TIFA). Funding of the project was through two TB Commitment Grants [0011-0549-1024 and 0011-0549-1025, received by ICMR-NIE] from JSI research & Training Institute, Inc. There was no additional external funding received for this study. The contents of this study document are the authors' sole responsibility and do not necessarily reflect the views of USAID or the United States Government or the institutions the authors are affiliated to.

**Competing interests:** The authors declare no conflicts of interest.

engagement of all stakeholders, adopting a community friendly approach, use of rapid diagnostic tests and digitalization were the perceived enablers. In some states ACF was implemented in general population (not restricted to high-risk population) following directives at state level. There were limited mechanisms to ensure ACF quality indicators were met before disbursing incentives and cross-verification of the aggregate ACF care cascade numbers that were reported in *Ni-kshay* (electronic TB information management system under NTEP). In addition to the state and district level implementers having limited understanding of concepts around ACF (quality indicators, number needed to screen and yield), we also inferred the presence of a 'know-do' gap for many activities under ACF. The suggested solutions were around capacity building and quality improvement strategies.

## Conclusion

The existing national ACF guidance should be revised to emphasize capacity building, need to carry out ACF in high-risk (not general) population, quality control-linked incentives, and regular implementation monitoring of the activities. This should contribute towards better coverage and improved quality translating into better ACF outcomes.

## Introduction

Globally, India has the highest tuberculosis (TB) burden and is one of the eight countries accounting for over two-thirds of the annual incident estimated people with TB [1]. Many people with TB are still missed (gap between the number of estimated and notified): 3.1 million globally and 0.4 million in India [1]. For this, the World Health Organization (WHO) recommends active case finding (ACF) in high-risk populations that have at least 0.5% of undetected TB [2]. These high-risk populations are also called marginalised and vulnerable populations. Under ACF, systematic screening for active TB is done outside of health care facilities. This includes institutional- and community-based systematic screening of high-risk populations as well as household contact investigation [3].

During 2018–19, a qualitative systematic enquiry was conducted among key stakeholders in high-burden countries, including Nepal and Vietnam, regarding factors influencing ACF policy development and implementation. Experts perceived ACF as a "double-edged sword" wherein the perceived benefits of ACF were linked to early diagnosis and treatment, and perceived harms were related to inappropriate design and implementation [4, 5]. The experts believed that increasing the knowledge and awareness of TB among patients and communities, providing training for the health staff, and increasing the monetary incentives will pave the way for scaling up ACF [6, 7].

During 2013–15, under The Global Fund-supported project *Axshya* (*Axshya* means without TB in Sanskrit) in India, ACF among high-risk populations improved case notification rates at the TB unit level compared to TB units implementing passive case finding (PCF) only [8]. During 2016–17, under the same project, ACF reduced health-system level diagnosis delay at the national level. ACF-detected patients incurred lower costs and lower chances of catastrophic costs during TB diagnosis than PCF-detected patients. ACF-detected patients did not have significantly better treatment outcomes when compared to PCF-detected patients [9–11]. One of the limitations was the lack of qualitative systematic enquiry to explore the provider and beneficiary perspectives for the above findings [12].

The Government of India's commitment to achieve Sustainable Development Goals' TB targets suffered a setback due to the emergence of the COVID-19 pandemic [13, 14]. When compared to 2019, the TB case notifications fell by 25% in 2020 [15]. As a part of COVID-19 mitigation measures, starting July 2021, there was a push to implement ACF among high-risk populations [16].

In 2022, India's National TB Elimination Program (NTEP) commissioned a national level evaluation of ACF (TB ACF evaluation project, August 2022 to February 2024) to guide evidence-based strategic planning. Under the first phase of this evaluation project, we assessed the frequency, scale (coverage) and quality of ACF at district, state and national level for the year 2021 using routinely collected secondary ACF data in *Ni-kshay* (electronic TB information management system under NTEP) [16]. Data on high-risk populations and presumptive TB detected were not consistently available to calculate all the ACF quality indicators (see Fig 1) [17]. Based on the data available, three derived ACF scale and quality indicators were used (see Table 1) [16]. We observed that comprehensive mapping of the high-risk populations in the districts was not done at the beginning of the year. Most states implemented one ACF cycle (that is intended to screen the high-risk populations once in the year), and none of the 36 states or 768 TB districts met all three (derived) ACF scale and quality indicator cut-offs. At national level, the number needed to screen was high (NNS = 2824) [16]. Our observation (anecdotal) was that the state and district level stakeholders implementing ACF need to be re-oriented towards the meaning of expected yield from (proportion of undetected TB in the high-risk population detected as a result of ACF) an ACF cycle, need for adequate planning with focus on reducing the losses between screening and testing, improving data quality and importance of monitoring ACF quality [16].

This called for systematic exploration of the NTEP staff's (provider) perspective through qualitative data collection methods. Most of the qualitative studies from India on ACF implementation before this were local with limited national relevance [18, 19]. Hence, this qualitative study (second phase of the evaluation project) was conducted (in 2023) to explore the 'why' (enablers and barriers) and 'what' (suggested solutions) of the above findings from the providers' perspective.

## Methods

### Ethics approval

This study was part of the second phase of the USAID/JSI-supported, ICMR-NIE-led TB ACF evaluation project. The evaluation project was approved by ICMR-NIE's ethics committee (NIE/IHEC/202201-10 dated 21 Mar 2022). Administrative approvals were taken before the conduct of this study. We obtained written informed consent from the study participants, and the ethics committee approved the consent process. To ensure confidentiality, soft copies of the transcripts and audio files were kept in a password-protected computer folder accessible only to the investigators analysing the data.

### Study design

This was a descriptive qualitative study. The theoretical framework underpinning this study was content analysis [20].

### Study setting

As of 2023, the NTEP infrastructure includes central TB division, state TB cells (n = 36), district TB cells (n = 768), sub-district (block) level administrative TB units (n≈6700) and

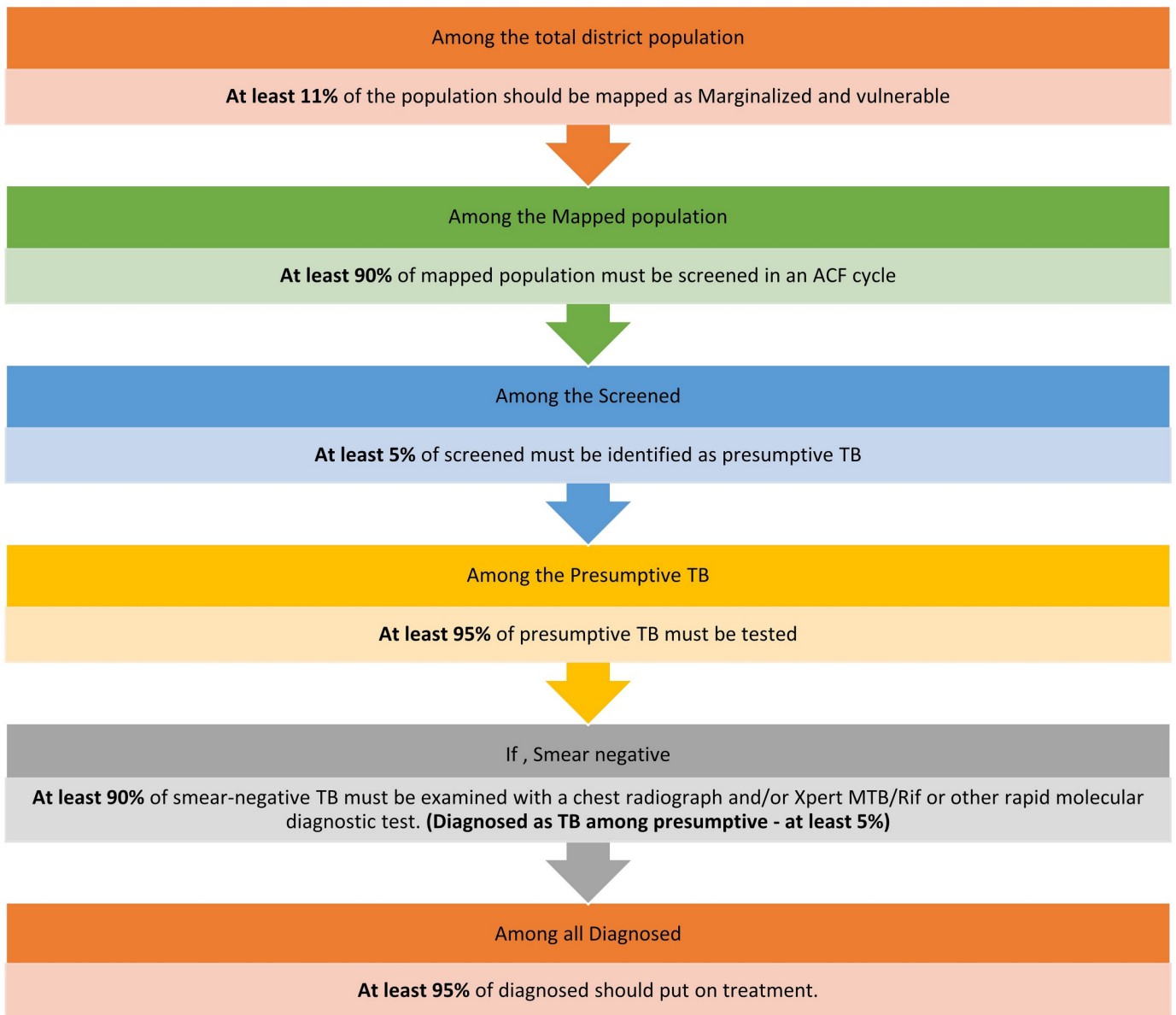

**Fig 1. Scale and quality indicators\* for an ACF cycle for TB, India as per the national ACF guidance 2019 [17].** ACF: active case-finding; TB: tuberculosis; Reprinted with permission from Shewade HD et al [16] under a CC BY license; first two indicators are ACF scale indicators and the rest are ACF quality indicators.

peripheral health institutions (public and private) (n≈2,70,000) that provide TB diagnosis and treatment services [21, 22]. Each district under the NTEP has a district TB officer (medical), a dedicated district program coordinator (nonmedical), a data entry operator and other sub-district level staff.

In 2017, the central TB division guided all state TB cells to carry out ACF thrice a year (January, July and December) among high-risk groups (three ACF cycles) [23]. The same guidance was updated in 2019 [17]. One ACF cycle is the dedicated period the district identifies when ACF will be done with the plan to cover all the district's high-risk populations [16]. Based on feasibility and considering resource constraints, many states divided each ACF cycle into

Table 1. The three derived^ ACF scale and quality indicators for an ACF cycle and their cut-offs used in the first phase of TB ACF evaluation project [16].

| Derived TB ACF indicators | Derived cut-off used in first phase (secondary data analysis for the year 2021) [16] |
|---|---|
| Percentage screened among the district's population (ACF scale indicator) | **At least 10%** (at least 11% of the district population to be mapped as high-risk population * at least 90% of the mapped population should be screened = 10%) |
| Percentage tested among screened (ACF quality indicator) | **At least 4.75%** (at least 5% of screened to be identified as presumptive TB * at least 95% of presumptive should be tested = 4.75%) |
| Percentage diagnosed (microbiologically / clinically) among tested (ACF quality indicator) | At least 5% (no change) |

TB-tuberculosis; ACF-active case finding;

^data on high-risk populations and presumptive TB detected were not consistently available across districts to calculate all the ACF quality indicators as recommended by the program and shown in Fig 1 [16]

multiple ACF rounds; some even implemented ACF on fixed days in a month throughout the year, while some focussed only on specific high-risk groups.

Mapping all high-risk populations is crucial for an efficient ACF campaign. It involves identifying and enumerating all the high-risk populations in the district and adding to *Ni-kshay*. Against each mapped population, aggregate numbers of the ACF care cascade are filled for each ACF activity day. *Ni-kshay* did not have the provision of providing the date range of each ACF cycle if a district planned to implement more than one ACF cycle in a year. At various levels, this information is essential to aggregate the ACF care cascade data of each ACF activity day against one ACF cycle (Fig 1).

The 2017 ACF guidance recommended the use of a paper-based ACF tool in the field to collect individual level information (name, age, gender and contact number) of those screened, followed by details on sputum collection, testing and diagnosis if identified as presumptive TB [23]. The 2019 guidance did not include this [16, 17]. Use of paper-based ACF tool in the field enables cross-checking of the ACF care cascade related aggregate numbers that are reported in *Ni-kshay*.

## Study population

Under the simultaneously implemented third phase (January to December 2023) of the ACF evaluation project, in 30 randomly sampled NTEP districts from nine states, we prospectively collected data to determine the losses in ACF care cascade, extent of use of Xpert MTB/Rif© and Truenat (both are nucleic acid amplification tests—NAAT) upfront among ACF-detected presumptive TB and compare the disease severity at diagnosis and pre-treatment delays among ACF- and PCF-detected patients. Project research assistants were posted in these study districts (one per district) to facilitate third phase-related data collection.

These 30 NTEP districts from nine states formed a convenient, accessible sample for the qualitative systematic enquiry (second phase of the project). Among these 30, six districts (from six different states) were purposively chosen: Ahmedabad Municipal Corporation (Gujarat), Jaipur I (Rajasthan), Aurangabad (Bihar), East Khasi Hills (Meghalaya), Deogarh (Odisha), Pudukkottai (Tamil Nadu) (see Fig 2). Data collection was done from the districts as well as the corresponding state (by visiting the state capital city). The purpose of including these districts varied district-by-district and are listed below i) high proportion of population screened and a high proportion of diagnosis among tested ii) low proportion tested among

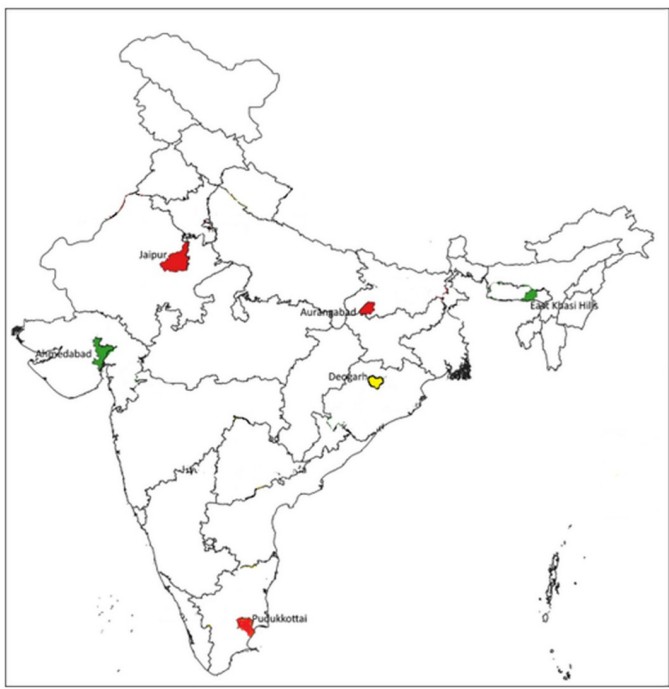

**Fig 2. State map of India showing the six districts selected for qualitative data collection, February to August 2023.**

screened and low proportion diagnosed among tested iii) no ACF activity or cut-off not met for all the three ACF scale and quality indicators (Table 1) iv) issues in transcribing ACF data into *Ni-kshay* portal v) ACF implementation in a metropolitan urban city vi) ACF implementation in hilly areas.

In these districts (and the corresponding states), between February (21-02-2023) and August 2023 (10-08-2023), after brainstorming, we included key NTEP managers and implementers who were experienced, vocal and perceived to provide insights into TB ACF planning, implementation, and monitoring. We included state-level program managers (state TB officers, WHO NTEP medical consultants based in the state capital), district-level managers (district TB officers, WHO NTEP medical consultants supporting the study district) and sub-district level implementers (the senior treatment supervisors (STS), TB-health visitors (TB-HVs) in urban areas). We also included the project research assistants posted in the districts. We realised that while collecting third phase-related data, they had gathered significant insight into ongoing ACF. Additionally, state-level program managers from two states (outside the convenient sample of nine states from phase three of the project) from north-eastern India were also included due to good performance in 2021 [16]. There were no refusals to participate. Eventual sample size was guided by data saturation.

## Data collection

The data collection method was one-to-one key informant interviews (KII). These were conducted and audio recorded by an investigator (PR) trained and experienced in qualitative research. Data was collected after obtaining permission and written informed consent to participate in the study. No incentives were given to participate in the study. Participants were approached over the phone first and then over email.

Participants were explained the purpose of the study and informed about the findings of the first phase of the project using infographics and tables. An interview guide with broad, open-ended questions and probes was finalised based on these findings. Interviews were conducted in English and the local language. Audio recording (after consent) and verbatim notes were taken during the interviews. After the interview, the summary of the discussions was read back to the participants to ensure participant validation. Additionally, an investigator (HDS) reviewed the audiotapes to reduce bias and increase interpretive credibility.

Interviews from February to May 2023 (in person) were conducted at the date, time, and place convenient to the participants. Only the participant and investigator were present during the interview. In addition, between June and August 2023, repeat online video interviews (there were no refusals) were conducted to dig deeper into what emerged and clarify conflicting ideas from earlier interviews. During this period, online video interviews were also conducted with experienced WHO NTEP medical consultants from two states in north-east India and project research assistants. Only the participant and investigator were present during the online video interviews.

The investigators involved in data collection worked in a national public medical research institute and were well-versed with the NTEP. The investigator who conducted the interviews (PR) was not part of NTEP (external member). She was responsible for overall management of the TB ACF evaluation project that included human resource management (includes project research assistants). The qualitative data collected from project research assistants was about ACF implementation by NTEP and not about their own performance (regarding phase three project related data collection). Hence, we do not think that this relationship between investigator and project research assistants would have affected the richness of data.

## Data management and analysis

Based on the audio records and notes, research interns (n = 6) translated and transcribed all the interviews verbatim within one week of data collection (under the guidance of PR). This was subsequently validated by two investigators (PR and SKP). These two investigators, trained in qualitative research, read the transcripts to become familiar with the data and they independently did a manual descriptive thematic analysis of the transcripts. Thematic analysis is appropriate for determining solutions to real world problems [24]. Using inductive reasoning, the investigators condensed the raw narrative data into themes and categories based on valid inference and interpretation. Data were coded using the Microsoft Word 2010 (Microsoft, Redmond, WA, USA) comment feature. The decision on coding rules and theme generation were done a priori by using standard procedures and in consensus. The themes emerged (not predetermined) by collating together the results of the coding process. Similar themes were further clubbed together to form broad categories. A third experienced investigator (HDS) reviewed the codes, themes and categories to reduce subjective bias and increase interpretive credibility. The third investigator also discussed and resolved any inconsistencies in the codes and themes generated by the two investigators (PR and SKP). The final codes, themes and categories were related to the original data to ensure that the results reflected the data [25]. The results were shared with the stakeholders for their feedback and approval. The results were reported using the "Consolidated Criteria for Reporting Qualitative Research" (COREQ) guideline [26].

## Results

A total of 34 KIIs were conducted. Of these four were repeat interviews. Table 2 depicts the study participant characteristics and the duration of interviews. The average length of the interviews was 30 minutes.

**Table 2. Study participant (NTEP staff) characteristics and duration of one-to-one interviews in selected districts/states of India, 2023.**

| One-to-one interview type | Designation (n) | Demographic details | Experience in that position | Duration of interview in minutes |
|---|---|---|---|---|
| **Key informant interviews—in person** | State TB officer (n = 2) | 64/M | 6 months as charge | 20 |
| | | 62/M | 4 years | 40 |
| | STDC additional director (n = 2) | 52/M | 2 years 6 months | 30* |
| | | 60/M | 5 years | 30 |
| | WHO NTEP consultants based in the capital city (n = 3) | 45/F | 6 years | 40** |
| | | 48/M | 4 years | 32 |
| | | 49/M | 14 years | 40 |
| | WHO NTEP consultants in charge of study districts (n = 4) | 40/M | 3 years | 16 |
| | | 38/F | 3 years | 50 |
| | | 50/M | 4 years | 27 |
| | | 37/F | 3 years | 47*** |
| | District TB officers (n = 6) | 45/M | 3 years | 15 |
| | | 60/M | 5 years | 36 |
| | | 62/M | 6 years | 33**** |
| | | 52/F | 4 years | 27 |
| | | 49/F | 2 years | 38 |
| | | 61/M | 6 years | 23 |
| | Senior treatment supervisors (n = 5) | 40/M | 10 years | 27 |
| | | 55/M | 9 years | 20 |
| | | 56/M | 6 years | 19 |
| | | 35/M | 6 years | 19 |
| | | 37/M | 8 years | 16 |
| | TB health visitor (n = 1) | 40/F | 5 years | 15 |
| **Key informant interviews—online video call** | WHO NTEP consultants based in capital city (n = 2) | 34/M | 3 years | 20 |
| | | 45/M | 6 years | 15 |
| | Project research assistants (n = 5) | 26/M | 1 year | 38 |
| | | 26/F | 1 year | 39 |
| | | 27/M | 1 year | 35 |
| | | 25/M | 1 year | 35 |
| | | 26/F | 1 year | 37 |

TB- tuberculosis; NTEP- national TB elimination program; STDC-state TB training and demonstration centre; WHO- world health organization;

*repeat interview for 24 minutes;

**repeat interview for 32 minutes;

***repeat interview for 36 minutes;

****repeat interview for 21 minutes (all repeat interviews were through online video calls)

We present the results in three separate parts: perceived enablers, perceived barriers and then suggested solutions.

## Perceived enablers

We identified six themes that were broadly classified into three major categories, operational planning, stakeholder engagement and technological advancements (Fig 3).

**Operational planning.** Two themes were categorized here, 'adequate budgets for ACF in PIP (program implementation plan) including incentives' and 'ACF performance review mechanism'.

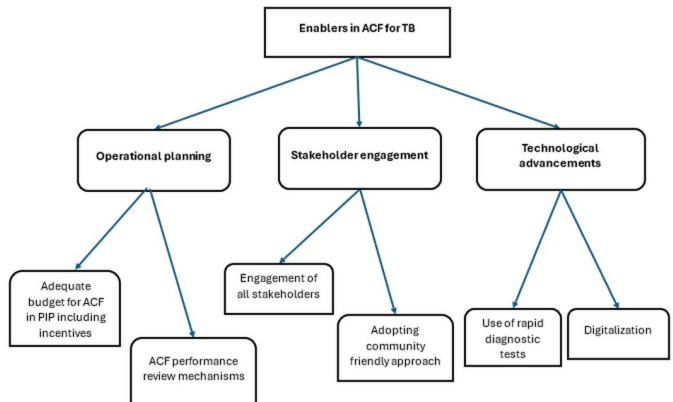

**Fig 3. Thematic analysis showing enablers as perceived by NTEP staff in implementing ACF for TB among high-risk populations in India (2023).** TB-tuberculosis; ACF-active case finding; NTEP- national TB elimination program; PIP- program implementation plan.

'Adequate budgets for ACF including incentives' are required for furnishing the costs associated with human resources (training, incentives), consumables, vehicle hiring, health promotion materials, advocacy communication social mobilization activities, procuring cartridges or chips for rapid molecular tests, falcon tubes and printing of paper-based forms. Most of the states and districts preplanned and requested budgets in PIPs for smooth implementation of ACF cycles.

The states adopted various 'ACF performance review mechanisms' and suggested feedback for improvement to the districts.

*"Considering the importance of the ACF activities, state NHM [National Health Mission] is reviewing it on routine weekly basis. . .*

*(NTEP consultant based at state capitWHO NTEP consultant based at state capital, 14 years' experience)*

*"We have particularly analyzed it, NNS [number needed to screen] through microscopy and NNS through NAAT [nucleic acid amplification test] and reviewed by state NHM [National Health Mission] also. . ."*

*(WHO NTEP consultant, three years' experience)*

**Stakeholder engagement.** Two themes were categorized here, 'engagement of all stakeholders' and 'adopting a community friendly approach'.

One of the important themes identified was 'engagement of all stakeholders' during IEC [information education communication] and ACSM [advocacy communication and social mobilization]. Some states engaged district collectors and local political leaders to launch ACF activities as it results in strong political commitment, advocacy, and visibility.

*"First thing is only STS and TB-HV goes to conduct ACF. . . . . . . Along with them, if other community influencers go for ACF, it is more efficient."*

*(Project research assistant posted at study district, one year experience)*

The field workers involved in ACF activity try to 'adopt community friendly approach'. Also, the NTEP managers identified that conducting ACF early in the morning, evenings, holidays, weekends resulted in better yield and reaching people with presumptive TB. Some states conducted integrated ACF campaign where in TB screening happened along with other disease screening.

*"We learnt it [need to visit on holidays and early morning and evenings] over period."*

*(WHO NTEP consultant based at state capital, six years' experience)*

**Technological advancements.** Two out of six themes were categorized here, 'use of rapid diagnostic tests' and 'digitalization'. Most of the stakeholders agreed with the fact that use of rapid diagnostic tests like Xpert MTB/RIF$^{©}$ and Truenat upfront increases sensitivity for the diagnosis of TB and has the potential to increase the yield of TB during ACF activities. Many states have taken steps to double the number of rapid diagnostic machines.

As per NTEP managers 'digitalization' like use of digital X-rays in TB diagnosis (one state), and use of digital mobile application (one state) instead of using paper-based ACF tool in the field made ACF activities more feasible and easier to monitor.

While we explored the enablers, we understood that in some states ACF was implemented across the entire population (not restricted to high-risk population) following directives at state level. Though the WHO NTEP consultants had a fair idea of ACF quality indicators, NNS and yield, the state and district level implementers had limited understanding of the same. Despite the existence of incentives in the PIP, the coverage of disbursement of the same remained suboptimal. Additionally, there were limited mechanisms in place to verify the aggregate ACF care cascade numbers (expected to be derived for an ACF activity day from the paper-based ACF tools used in the field) reported in *Ni-kshay* and whether ACF quality indicators were met before disbursing incentives. Another challenge was a lack of community trust in the health care system. We delved deeper to understand these challenges and to uncover the reasons for the observed gaps during the first phase.

## Perceived barriers

We identified 11 themes broadly classified into four major categories, 'administrative', 'logistic', 'technical' and 'socio-cultural and geographic'.

**Administrative barriers.** Three out of 11 themes were categorized here, 'heavy work burden', 'inconsistencies in ACF cycles', and 'lack of supervision and monitoring'. One of the barriers that every stakeholder consistently put forward was the 'heavy work burden', especially among ASHA workers [Accredited Social Health Activist, community-based volunteers who received incentives under various health programs]. Many expressed concern that ASHAs working on a target-based approach may compromise the quality of ACF.

*"ASHA is doing a lot of work, and I have also seen in few districts that they were so pressurised that they brought their own sample [specimen]. . ."*

*(WHO NTEP consultant, three years' experience)*

Inconsistencies in the number of ACF cycles across different districts within a state every year was felt by most of the stakeholders. Some of them cited that it was feasible to implement three ACF cycles per year, whereas some cited that even one ACF cycle was not possible. With the lack of capture of date range of one ACF cycle in a district in *Ni-kshay* and these variations

in number of ACF cycles made the routine monitoring of scale and quality (see Fig 1) a humongous task at the state and national level. Additionally, in some states, ACF activities within a district were not implemented in a planned manner (with adequate microplanning) but on an ad-hoc basis.

Though some NTEP staff quoted state level review mechanisms as an enabler, others reported a lack of stringent monitoring of ACF activities.

*"[Lack of] Monitoring is the main barrier because in active case finding, until we are not able to monitor the ACF activity at the field level, we are not able to get a good yield..."*

*(WHO NTEP consultant, three years' experience)*

**Logistic barriers.** Four out of 11 themes were categorized here, 'financial issues', 'issues in diagnostic services', 'lack of human resources' and 'issues in specimen quality, collection and transport'. A common barrier was 'financial issues' related to the lack of timely incentives for ASHA workers. This would make them lose interest and eventually work for the program which provides them with higher and timely incentives.

*"Earlier, it was said that Rs 100 would be paid to each ASHA worker to cover 50 households, but the money was not received..."*

*(Senior treatment supervisor, six years' experience)*

'Issues in diagnostic services' such as non-availability of cartridges and chips was experienced by many staff. This contributed to diagnosis delay and attrition in the ACF care cascade mechanism. A common barrier identified by almost all NTEP staff was 'lack of human resources.'

*"The problem is that we don't have efficient volunteers, and the work cannot be done effectively. The HR [human resource] position of NTEP staff itself is vacant at our place, and at some levels, we don't have STS, and some other positions are also vacant. so, we are not able to cover ACF at some places due to lack of human resources..."*

*(STDC Additional director, five years' experience)*

Many NTEP managers identified challenges in 'specimen quality, collection, and transport' as a significant barrier. They pointed out the absence of a courier system in their districts and states leading to diagnostic delays.

*"We have a courier facility for sputum collection from CHC [community health centre] to DTC [district TB centre] and DTC to State TB cells, but there is no courier system from the health and wellness centre to CHC, where the microscopy is done."*

*(WHO NTEP consultant, four years' experience)*

**Technical barriers.** Two out of 11 themes were categorized here, 'issues in *Ni-kshay*' and 'existence of knowledge gaps'. Some NTEP staffs raised concerns about 'issues with *Ni-kshay*' particularly related to lack of data entry. They observed that field staff often prefer [due to familiarity] using google sheets to report ACF data when asked by the state over *Ni-kshay*.

Additionally, they noted lack of training on how to enter ACF related data in *Ni-kshay* and they suffer from increased workload in managing both hard and soft copies.

> *"We are trying to get data from Ni-kshay; not all districts are entering all the complete data. We have a paper-based [ACF] tool [for use in the field], and nobody reports in Ni-kshay. . ."*
>
> *(WHO NTEP consultant, three years' experience)*
>
> *"Currently, the usage of active case finding module is not very optimal in Ni-kshay as compared to the other modules, because other modules are on a real-time basis and ACF module is not. . ."*
>
> *(WHO NTEP consultant based at state capital, six years' experience)*

When we enquired stating that the *Ni-kshay* ACF module is not for use in the field but to capture aggregate numbers from ACF care cascade and that these numbers should be based on a paper-based ACF tool used in the field, we understood that some states were not using any paper-based ACF tool in the field.

> *". . ...no such rule to implement any paper-based tool [in the field]. They told just go to household, screen and if you find presumptive TB, just we have to bring that person to PHI [peripheral health institute]"*
>
> *(project research assistant)*

'Existence of knowledge gaps' was detected in a few but not consistently seen among many staff. The respondents expressed dissatisfaction with the training provided. Though they knew the importance of entering ACF data into the *Ni-kshay* portal, they cited lack of training as a barrier. Knowledge gap was observed in sub-district level staff regarding mapping of high-risk population.

> *"The state has not given training. Only an online session was provided. . ."*
>
> (*project research assistant*)
>
> *"In many places they have not [stressed the word 'not'] done the mapping properly. We have to map it na where you have to do. You can cover the whole general population, but you need to know the vulnerable population in your area."*
>
> *(WHO NTEP consultant, three years' experience)*

While exploring the reasons for knowledge gap, one stakeholder emphasized.

> *"See, many things are there in the system, which is doable, which is not being done, that's my answer. . ."*
>
> *(WHO NTEP consultant, three years' experience)*

**Socio-cultural and geographical barriers.** Two out of 11 themes were categorized here, 'existence of taboos and stigma' and 'issues in coverage'. To begin with, the taboos and stigma existed among specific population groups. Many of the field staff noticed that there exists a

negative perception or attitude among people, leading to restriction of family members from participating in ACF.

Another significant barrier identified was 'issues in coverage'. Field workers conducting ACF activities face difficulties like hilly terrains, extremes of temperature, and vast surface areas, which reduce their performance. Some even mentioned that public transport was not available in specific locations and financial constraints even if transport was available.

## Suggested solutions

We identified seven themes, broadly classified under two categories, 'ACF quality improvement strategies' and 'capacity building strategies'.

**ACF quality improvement strategies.** We identified four themes under this category, 'need for an integrated approach', timely incentives', 'regular cross check of ACF data' and 'need for user-friendly *Ni-kshay*'.

To improve community engagement and to maximize the overall impact of ACF, numerous NTEP staff proposed the 'need for integrated approach' of combining diverse health programs. This approach is seen as potentially more effective in addressing various health issues.

*"Instead of going for one program, if we go it as integrated health system approach for all the diseases, there would be any better participation by the community. . ."*

*(WHO NTEP consultant based at state capital, six years' experience)*

The NTEP managers highlighted that providing regular 'timely incentives' to ASHAs will motivate them.

*"CTD [central TB division] has not given any proper guidelines that we can give 5 rs [Indian Rupee], 10rs for one house, and each state is designing their own structure. They can give certain directions to the state government, then the improvement will be there. . ."*

*(district TB officer, three years' experience)*

The NTEP staff suggested approaches to 'regularly cross check ACF data'. These include feedback mechanism related to data entry, proposals for external quality monitoring and the significance of physical presence during ACF for effective monitoring.

*"Autonomous external quality monitoring mechanism can be set up with incentives, who are independent of the program. . ."*

(*WHO NTEP consultant based at state capital, six years' experience*)

*"During ACF, the only thing is to monitor them, it's like making your presence during the activity. . .."*

*(WHO NTEP consultant, three years' experience)*

Staff suggested the need for user friendly *Ni-kshay* portal. Some even discussed about the need for more flexibility and functionality in handling the data within the *Ni-kshay* portal.

*"The Ni-kshay application is not robust; this is a little complicated, so it should be user-friendly. . ."*

(*STDC Additional director, two years' experience*)

*"Regarding the ACF module with Ni-kshay, yes, definitely some sort of changes in terms of the way the data is captured, yeah, also if there can be an export of the data which is already been recorded…"*

*(WHO NTEP consultant based at state capital, six years' experience)*

**Capacity building strategies.**   We identified three themes under this category, 'laboratory strengthening', 'scaling up of resources' and 'ACF training modules for ASHAs'

The NTEP managers suggested optimal use of diagnostic services and the need to explore ways to improve its utilization. One STS with six years' experience emphasized the need for transport mechanism of sputum specimen from the field to PHCs. The necessity for 'scaling up of resources' for effective ACF activity was insisted by many stakeholders. Few of them suggested a solution to streamline data entry in field for ACF in a mobile application. They also highlighted on the importance of timely recruitment of staffs and volunteers.

*"If the ACF has some mobile application [field-based, like the one reported in one state], they can do the direct entry of the ACF, then we can get the [aggregate ACF] data directly from the field…"*

*(WHO NTEP consultant, three years' experience)*

The NTEP managers also insisted on the need for 'ACF training modules for ASHAs' to bridge the gap in training resources.

The providers' perspective on barriers and the suggested solutions to address them have been summarized in Fig 4.

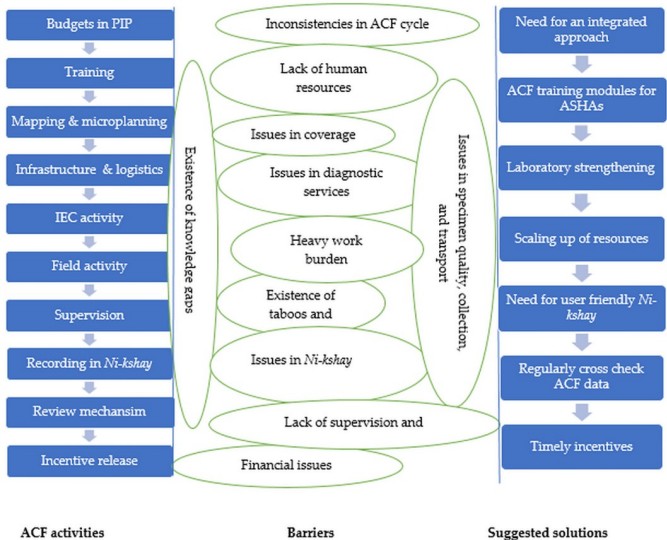

**Fig 4. Providers' perspectives into barriers and suggested solutions to improve ACF for TB among high-risk populations in India (2023), in relation to ACF activities.** TB-tuberculosis; ACF-active case finding; IEC-information education communication; PIP- program implementation plan.

## Discussion

In the context of TB ACF quality indicators being below the benchmark in India, this is the first ever multi-state qualitative study conducted with an effort to explore the planning, implementation, and monitoring of ACF from the perspective of state, district and sub-district level NTEP staff. Globally, community perspectives and policy level perspectives on ACF are well known and researched [4, 5, 27]. In our study, NTEP staff perceived that operational planning, involving community stakeholders, and technological advancements are crucial for successful ACF implementation. We also identified several barriers that impede effective ACF implementation. In some states ACF was implemented in general population (not restricted to high-risk population) following directives at state level. There were limited mechanisms to ensure ACF quality indicators were met before disbursing incentives and cross-verification of the aggregate ACF care cascade numbers that were reported in *Ni-kshay*. The state and district level implementers had limited understanding of concepts around ACF scale and quality indicators, number needed to screen and yield. In line with our findings, studies conducted elsewhere in India (Bihar, Karnataka), southeast Asia and other low- and middle-income countries have also reported similar challenges [18, 19, 28–31]. This suggests that these barriers to ACF implementation are faced not only by India but also by other low- and middle- income countries.

To address the identified barriers and enhance the yield, several key strategies were suggested by the NTEP staff. Below we have summarized the discussion in two parts. First, we begin with our inferences summarized in the form of 'know-do' gaps (knowledge-practice gaps). Then we present our recommendations in line with our findings.

To describe the 'know-do' gaps (see Table 3), we broadly categorized the ACF activities into three major headings: planning (at district/state level), implementation (sub-district level) and monitoring and evaluation (at district/state level) [17].

There was a knowledge gap for the following four activities: mapping the high-risk groups (implementation, sub-district level), microplanning in these mapped areas (implementation, sub-district level), mechanisms to cross-check aggregate ACF care cascade data reported in *Ni-kshay* (monitoring and evaluation, state/district level) with data in ACF tool used in the field (if any), and quality-control based timely release of incentives (monitoring and evaluation, state/district level). The reason for the existence of knowledge gap among district and subdistrict level implementers was lack of adequate training. Regarding the use of ACF tool in the field, while it was not recommended in the updated 2019 ACF guidance, it was recommended in the 2017 guidance [17, 23]. A systematic review on ACF among homeless documented that the strongest evidence for improving uptake of screening was for incentives [32]. However, incentives need to be provided subject to cross-checking in ACF tool and good quality ACF activity.

Except for budget estimation in PIP, conducting IEC activities and consolidation of reports, we found significant 'know-do' gap in other activities. Under planning (state/district level), though there was theoretical knowledge on ACF concepts and principles at state/district level, we found a notable gap in transfer of this knowledge to the sub-district level during trainings. Poor training had an impact on major part of ACF implementation. Though most of the states knew about the use of ACF tool in the field, it was not consistently used for data collection and monitoring. This was further complicated by implementation of ACF on an *ad hoc* basis in some states.

In addition, considering the perceived complexity in *Ni-kshay* ACF module, a parallel reporting mechanism in the form of google sheets was utilized in some of the states. In *Ni-kshay*, ACF data (aggregate numbers of ACF care cascade) is captured for each ACF activity day against a mapped population identifier. There could be multiple ACF activity days against

**Table 3. Investigator inferences on the 'know-do' gap across various steps in ACF for TB among NTEP staff in India, 2023.**

| S. No | Steps in ACF for TB* | Know | Do |
|---|---|---|---|
| 1 | **Planning (state / district level)** | | |
| | State, district and block-level training of NTEP staff regarding knowledge about ACF concept and principles** and ACF implementation/monitoring | Yes | No |
| | Estimation of budgets, IEC materials and logistics for the state and districts | Yes | Yes |
| 2 | **Implementation (sub-district level)** | | |
| | Mapping of the high-risk population | No | No |
| | Microplanning of activities in mapped areas | No | No |
| | IEC activity along with ACF | Yes | Yes |
| | ACF tool (paper-based or web app) used in the field during ACF activity | Yes | No |
| | Electronic data reporting (*Ni-kshay*) | Yes | No |
| 3 | **Monitoring and evaluation (state/district level)** | | |
| | Supervision of field activity | Yes | No |
| | Mechanisms to cross-check data in ACF tool used in the field (if any) and aggregate ACF care cascade data reported in *Ni-kshay* | No | No |
| | Quality-control based, timely release of incentives | No | No |
| | Consolidation of report & organizing meeting for review | Yes | Yes |

TB-tuberculosis; ACF-active case finding; NTEP- national TB elimination program; IEC-information education communication;

*in line with NTEP ACF guidance document 2019 [17];

**what is one ACF cycle, type of population where ACF should be done and how to assess the quality and yield of ACF for one ACF cycle or round?

a mapped population identifier. There could be multiple mapped population identifiers in a TB unit. Hence to generate ACF quality indicators for an ACF cycle / round, district and sub-district level implementers will have to append the ACF activity data and merge it with mapping identifier data. Many implementers may not be having the skills to do this. Hence, to simplify things, it appears google sheet was used.

We suggest some recommendations (six tips) which are in line with those suggested by the stakeholders and based on investigators' experience in India's NTEP. These six tips can guide revisions in national ACF guidance [17, 23].

First, planning is essential for estimating the resource availability and allocating budgets for context-specific ACF. In addition to ongoing steps to fill vacant NTEP posts, a list of volunteers should be prepared for their services in ongoing ACF cycles. Second, regular training for staff at various levels (separate training modules by type of staff) must be conducted with focus on the knowledge gaps.

Third, incentives should be granted to field staff or volunteers for each household visited and for every specimen collection and transport to the laboratory. Over a pre-decided period (required for sufficient sample size to assess test positivity), all criteria need to be fulfilled before the release of these incentives (see Fig 5). These include i) submission of completed ACF tool used in the field (from screening to test results) to the nearest laboratory, ii) random cross-checks of a specified percentage of those screened and identified as presumptive TB, iii) cross-checking of all ACF-detected TB (that they were detected as a result of ACF and not merely detected during the ACF period) and iv) confirmation that at least 5% (a lower cut-off may be considered) of the tested samples are positive for TB (microbiological or clinically

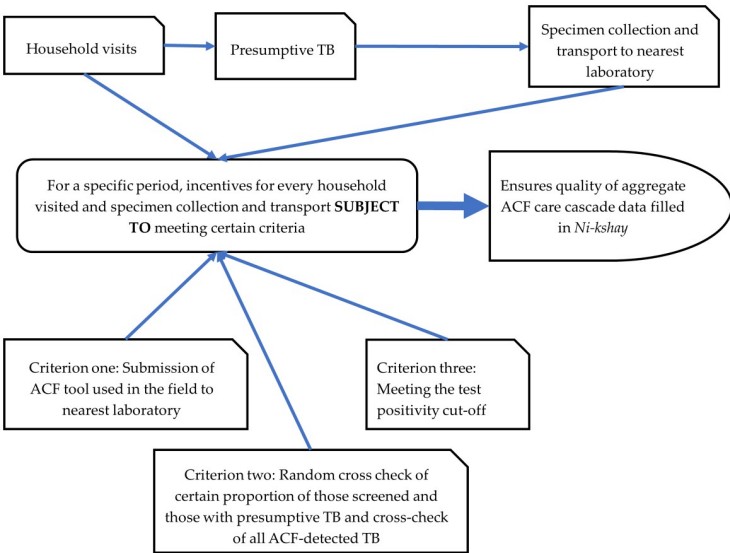

**Fig 5. Proposed mechanism for quality-control based, timely release of incentives for field staff and volunteers involved in ACF for TB in India.** TB-tuberculosis; ACF- active case finding.

confirmed). This approach circumvents the need for a courier system (see Fig 5). In areas where the distance between the field and the laboratory is significant (say hilly terrain), an intermediate location may be designated for specimen deposition, followed by using a human courier system. This approach may also address the issue of low proportion of testing among screened populations (more in population-based high-risk groups when compared to institutional high-risk group) which fared the worst among the ACF quality indicators in 2021 [16].

Fourth, simplifying the ACF module in *Ni-kshay* portal and making it more user-friendly could encourage better utilization by sub-district level STS and district TB officer. In *Ni-kshay*, there must be provisions to: i) update population numbers against each type of high-risk groups in a district (mapping) at the beginning of the year ii) add date range for each ACF cycle (if the district plans more than one cycle) in the year iii) enter multiple records against each month in an ACF cycle or year, where against each record, the type of high-risk group covered (dropdown option) and the number screened, presumptive, tested and diagnosed are entered, and iv) export (overall and high-risk group wise) the following aggregate numbers against each month in an ACF cycle: mapped population type, number screened, number presumptive, number tested and number diagnosed.

Fifth, establishing mechanisms for regular cross-checking of ACF activities and ACF data reported in Ni-kshay (aggregated from ACF tool used in the field) is essential. This includes regular feedback for data entry, proposals for external quality monitoring, and emphasizing the importance of physical presence/supervision during ACF. Findings from a qualitative study in Ghana further support our observations, indicating that consistent visits and the knowledge of being monitored act as motivation in tuberculosis case detection [29]. Providing timely feedback and taking corrective actions at frequent intervals may help in course correction.

Finally, as reported from Zambia and China, an integrated approach that combines TB ACF with various other health programs could reduce the stigma and improve community participation, where they become more engaged and receptive [33, 34].

These six tips appear to be in line with the 'anecdotal' recommendations made in our previous research paper from phase one of the project [16]. The national ACF guidance should be modified keeping these in mind. International and national guidance and framework for implementing ACF for TB are available [2, 17, 23, 31]. These lack specifics regarding operationalizing ACF in the context of an ACF cycle, ensuring scale and quality. Hence, there is scope for developing practical and actionable ACF practice document for use by state and district TB officers, after considering modifications based on local context.

There was one limitation. Though the sampling was purposive, most of it was from a convenient sample of 30 districts from nine states that were part of the simultaneously conducted third phase of the project. Due to the presence of our project research assistants in these study districts and owing to orientation meetings held for the district and state stakeholders, there could be masking of the knowledge gaps on the concept and principles of ACF among NTEP staff. Hence, the knowledge levels explored here could be a best-case scenario.

## Conclusion

In 2021, we observed that the frequency, scale and quality of ACF for TB was suboptimal in India with major gaps in planning, mapping of high-risk population and extent of testing among those screened [16]. With this background in mind, this qualitative study explored the enablers, barriers, and suggested solutions to improve ACF for TB from the NTEP staffs' (provider) perspective. We inferred the presence of a 'know-do' gap for many activities under ACF. A comprehensive and collaborative approach that addresses administrative, logistic, technical, and socio-cultural barriers identified in this study is necessary. Working on the suggested six tips provided by us (planning for context specific ACF, training to address the identified know-do gaps, quality-control linked incentives, simplification of ACF module in *Nikshay*, mechanisms for regular cross-checking of ACF activities and data and TB ACF combined with other disease screening initiatives), the existing national ACF guidance should be revised. This will contribute towards better scale, quality, yield and eventually ACF outcomes (improved detection, early detection, and better treatment outcomes) [2, 3]. The findings and recommendations from this study could be of interest and relevance to TB programme managers and policy makers not only from India, but also from other high TB burden countries.

## Acknowledgments

The authors sincerely thank the support and contribution received from all the state TB cells, district TB cells and the WHO NTEP medical consultant network in India. We also acknowledge the contribution by the interns of ICMR NIE, Chennai, India who provided data transcription support: Mr SM Aakash, Ms M Nisha, Ms Archita Govardhana, Ms Suhana Khatoon B, Ms Aishwarya Dhumale and Mr Mahesh Gomasa.

## Author Contributions

**Conceptualization:** Hemant Deepak Shewade, Prabhadevi Ravichandran, S. Kiran Pradeep, G. Kiruthika, Devika Shanmugasundaram, Joshua Chadwick, Swati Iyer, Amar N. Shah, Bhavin Vadera, Venkatesh Roddawar, Sanjay K. Mattoo, Kiran Rade, Raghuram Rao, Manoj V. Murhekar.

**Data curation:** Hemant Deepak Shewade, Prabhadevi Ravichandran, S. Kiran Pradeep.

**Formal analysis:** Hemant Deepak Shewade, Prabhadevi Ravichandran, S. Kiran Pradeep.

**Funding acquisition:** Hemant Deepak Shewade.

**Methodology:** Hemant Deepak Shewade, Prabhadevi Ravichandran, S. Kiran Pradeep, G. Kiruthika, Devika Shanmugasundaram, Joshua Chadwick, Swati Iyer, Aniket Chowdhury, Dheeraj Tumu, Amar N. Shah, Bhavin Vadera, Venkatesh Roddawar, Sanjay K. Mattoo, Kiran Rade, Manoj V. Murhekar.

**Project administration:** Hemant Deepak Shewade.

**Supervision:** Hemant Deepak Shewade, Kiran Rade, Raghuram Rao, Manoj V. Murhekar.

**Visualization:** Hemant Deepak Shewade, Prabhadevi Ravichandran, S. Kiran Pradeep, G. Kiruthika, Devika Shanmugasundaram, Joshua Chadwick, Aniket Chowdhury, Dheeraj Tumu, Amar N. Shah, Bhavin Vadera, Venkatesh Roddawar, Sanjay K. Mattoo, Kiran Rade, Raghuram Rao, Manoj V. Murhekar.

**Writing – original draft:** Hemant Deepak Shewade, Prabhadevi Ravichandran.

**Writing – review & editing:** Hemant Deepak Shewade, Prabhadevi Ravichandran, S. Kiran Pradeep, G. Kiruthika, Devika Shanmugasundaram, Joshua Chadwick, Swati Iyer, Aniket Chowdhury, Dheeraj Tumu, Amar N. Shah, Bhavin Vadera, Venkatesh Roddawar, Sanjay K. Mattoo, Kiran Rade, Raghuram Rao, Manoj V. Murhekar.

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
