## [Decision Letter · Decision Letter 0]

26 Mar 2024

PONE-D-23-43822Bridging the “know-do” gap to improve active case finding for tuberculosis in India: A qualitative exploration into program staffs’ perspectivesPLOS ONE

Dear Dr. Shewade,

Thank you for submitting your manuscript to PLOS ONE. After careful consideration, we feel that it has merit but does not fully meet PLOS ONE’s publication criteria as it currently stands. Therefore, we invite you to submit a revised version of the manuscript that addresses the points raised during the review process.

We look forward to receiving your revised manuscript.

Kind regards,

Ravi Ranjan Kumar, P.hd

Academic Editor

PLOS ONE

Journal Requirements:

This study was commissioned by India’s national TB elimination program, with funding support from USAID through John Snow International (JSI) under the TB Implementation Framework Agreement (TIFA). The contents of this study document are the authors' sole responsibility and do not necessarily reflect the views of USAID or the United States Government.

4. In the online submission form, you indicated that The anonymized transcripts are available on request to the corresponding author subject to signature of a data confidentiality agreement.

ICMR-NIE led USAID/JSI supported TB ACF evaluation project is a collaborative effort involving ICMR-National Institute of Epidemiology (ICMR-NIE), Chennai, India (lead); USAID India, New Delhi, India; JSI India, New Delhi, India; The WHO Country Office for India, New Delhi, India; and Central TB Division, Ministry of Health and Family Welfare, Government of India, New Delhi, India. The authors sincerely thank the support and contribution received from all the state TB cells, district TB cells and the WHO NTEP medical consultant network in India. We also acknowledge the contribution by the interns of ICMR NIE, Chennai, India who provided data transcription support: Mr SM Aakash, Ms M Nisha, Ms Archita Govardhana, Ms Suhana Khatoon B, Ms Aishwarya Dhumale and Mr Mahesh Gomasa. 

This study was commissioned by India’s national TB elimination program, with funding support from USAID through John Snow International (JSI) under the TB Implementation Framework Agreement (TIFA). The contents of this study document are the authors' sole responsibility and do not necessarily reflect the views of USAID or the United States Government.

Reviewers' comments:

Reviewer's Responses to Questions

**Comments to the Author**

1. Is the manuscript technically sound, and do the data support the conclusions?

Reviewer #1: Yes

Reviewer #2: Partly

2. Has the statistical analysis been performed appropriately and rigorously? 

Reviewer #1: Yes

Reviewer #2: N/A

3. Have the authors made all data underlying the findings in their manuscript fully available?

Reviewer #1: No

Reviewer #2: No

4. Is the manuscript presented in an intelligible fashion and written in standard English?

Reviewer #1: Yes

Reviewer #2: Yes

5. Review Comments to the Author

**Reviewer #1:** The manuscript, “Bridging the “know-do” gap to improve active case finding for tuberculosis in India: A qualitative exploration into program staffs’ perspectives” describes the enablers, barriers, and suggested solutions to improve TB active case finding India from NTEP staff (provider) perspective.

The authors have described well the introduction, and results. The methodology and the discussion need some improvement. The findings have public health importance to understand what factors may influence TB active case findings. The findings may also have importance in the context of infection prevention and control in community settings. However, the manuscript would require substantial revision to make it clearer and reader friendly.

The manuscript may be appropriate for this journal and could be considered for publication in PLOS ONE after addressing following issues.

Abstract:

The objective could be merged with the background section for clarity and conciseness. Methods:

Additional details are required in this section. The analysis section is very vague and warrants more information. This includes clarification on whether the interviews were recorded, how key informants were selected and why, the type of questions asked, who transcribed the recordings, whether the transcriptions were verbatim, how themes were identified, and if there were any pre-defined themes. Additionally, the resolution of inconsistencies between the two individuals involved in thematic analysis should be explained.

Results:

Clarify the term "Ni-kshay" as an international audience may not understand it. The conclusion regarding revising ACF guidance is too generic. Authors should specify the changes required based on the study findings.

Main manuscript-

Introduction- The statement “Based on the data available, we used three derived ACF quality indicators in our analysis (see Table 1) [16]. We observed that comprehensive mapping of the high-risk populations in the districts was not done at the beginning of the year.” sounds more like a description of the study methods and should be revised accordingly.

Study settings- Estimations regarding the number of sub-district (block) level administrative TB units and peripheral health institutions (public and private) providing TB diagnosis and treatment services should be provided. The study location, including the names of the districts and reasons for their selection, should be mentioned. Avoid unnecessary information.

Study population- Focus solely on describing the study population in this section.

“These 30 NTEP districts from nine states formed a convenient, accessible sample for the qualitative systematic enquiry (second phase of the project). Among these 30, six districts (from six different states) were purposively chosen: Ahmedabad Municipal Corporation (Gujarat), Jaipur I (Rajasthan), Aurangabad (Bihar), East Khasi Hills (Meghalaya), Deogarh 167 (Odisha), Pudukkottai (Tamil Nadu) (see Fig 2)” these should go under the study settings.

Data collection- Clarify how consent was obtained for the second round of interviews and whether it was planned in the study protocol. Explain the steps taken to maintain confidentiality and mention any refusals for the second round of interviews. Additionally, state the number of repeated interviews conducted.

Data management and analysis- Provide details regarding the number of people involved in data transcription and the average length of interviews. In Table 2, ensure the last column is labelled "Duration of interview in minutes."

Discussion- When referring to similar studies, mention the location to provide context for the audience. Interpret the findings rather than just describing them. Explain why knowledge gaps exist and provide reasoning for identified discrepancies.

**Reviewer #2:** The study is from a commissioned TB ACF evaluation project by India’s National TB Elimination (NTEP). India is trying to explore implementation of ACF, however, considering the costs and comparative insignificant case finding, passive case finding remained the cornerstone of NTEP for decades. The present study brings out some experiences based on qualitative research which is an important aspect. It is also important that authors addressed “know-do gap” in the ACF implementation.

1.The title should reflect “NTEP” program otherwise just stating program does not give an idea to readers.

2.Abstract: Objective should not include suggested solutions

3.34 key informant interviews should go as a part of methods.

4.Authors mentioned that they performed manual analysis, but in my opinion, it may sometimes be biased. Why did authors not use qualitative data management programs such as MAXQDA or Nvivo etc.? That would provide more objective analysis.

5.Lines 78-79: Authors mentioned “ACF did not have better outcome rates as compared to Passive case finding”. This needs to be interpreted carefully. Treatment outcomes do not necessarily always depend upon ACF and early diagnosis, but the entire treatment course as well. So, the comparison needs to be made considering various points in both methods. The rationale for undertaking qualitative research is not sufficiently stated.

6.The real problem with ACF implementation is whether incentives are going to continue as that would decide the sustainability of any program.

7.Methods section should be revised thoroughly as how actually they did the data management and analysis. Authors may describe how content analysis underpinned current research. Authors should describe the detailed method of thematic analysis.

Authors mentioned in the abstract that two investigators did a manual descriptive thematic analysis of the transcripts, independent of each other. A third investigator reviewed the same. It is unclear whether there were consistency or discrepancies in this method and how third person contributed. Authors may want to throw light on these issues.

8.Lines 218-220: “The data collected from project research assistants was about ACF implementation by NTEP and not about their own performance. Hence, we do not think that this relationship between investigator and project research assistants would have affected the richness of data”. This is not clear.

9.Ethics approval should be in the beginning of methods section.

10.In my opinion, some quantification from 34 interviews might help in some places to understand how relevant is that issue and its importance. Authors used “many of the” which is hard to follow.

11.The analysis needs to be organized. Authors mentioned numerous themes clubbed into a few categories. It is hard to follow.

12.The discussion should highlight that ACF strategies need to be implemented considering the ground situations which is often lacking in Indian context. Capacity building and quality improvement are very general suggestions as these are constant processes. In my opinion authors need to provide suggestions based on their own specific findings which will help to undertake focused interventions.

13.Lines 472-473: “In line with our findings, studies elsewhere have also reported similar

challenges [22–26]. Authors should first describe what other studies have reported and then discuss their findings whether they are confirming or having different observations.

14.The conclusion needs to be specific and based on findings rather than general.

6. PLOS authors have the option to publish the peer review history of their article (what does this mean?). If published, this will include your full peer review and any attached files.

Reviewer #1: **Yes: **Md Saiful Islam

Reviewer #2: No

---

## [Author Response · Author response to Decision Letter 0]

14 May 2024

Title: Bridging the “know-do” gap to improve active case finding for tuberculosis in India: A qualitative exploration into national tuberculosis elimination program staffs’ perspectives

[Kindly note a change in title based on a comment by reviewer]

PONE-D-23-43822

Article type: Research Article

Journal: PLOS One

Sub: Submission of revised manuscript

Dear Editor, 

We would like to express our sincere gratitude for the constructive comments from the reviewers. We found them very helpful for improving the quality of our manuscript. We have made every effort to address each comment as carefully as we can. All revised parts in the manuscript are in track change. Please find our point-by-point responses below. The line numbers mentioned in our response are the line numbers from the “Revised Manuscript with Track Changes”. 

We have also taken this opportunity to make some editorial changes for better ease of reading. If there are any further clarifications, we would be very glad to address them,

The correct updated funding statement is as follows

ICMR-NIE led USAID/JSI supported TB ACF evaluation project is a collaborative effort involving ICMR-National Institute of Epidemiology (ICMR-NIE), Chennai, India (lead); USAID India, New Delhi, India; JSI India, New Delhi, India; The WHO Country Office for India, New Delhi, India; and Central TB Division, Ministry of Health and Family Welfare, Government of India, New Delhi, India. This study was commissioned by India’s national TB elimination program, with funding support from USAID through John Snow International (JSI) under the TB Implementation Framework Agreement (TIFA). There was no additional external funding received for this study. The contents of this study document are the authors' sole responsibility and do not necessarily reflect the views of USAID or the United States Government.

With regards,

Dr Hemant D Shewade

Responses to Comments

Journal Requirements:

AUTHOR RESPONSE

Noted with thanks. We have ensured that manuscript meets PLOS ONE style requirements

Journal Requirements:

AUTHOR RESPONSE

We have removed funding information from manuscript file and ensured correct statement in the manuscript submission website and mentioned the same in the cover letter. We have removed funding related information any other parts of the manuscript. 

Journal Requirements:

This study was commissioned by India’s national TB elimination program, with funding support from USAID through John Snow International (JSI) under the TB Implementation Framework Agreement (TIFA). The contents of this study document are the authors' sole responsibility and do not necessarily reflect the views of USAID or the United States Government.

AUTHOR RESPONSE

We have removed funding information from manuscript file and ensured correct statement in the manuscript submission website and mentioned the same in the cover letter. We have removed funding related information any other parts of the manuscript. 

Journal Requirements:

4. In the online submission form, you indicated that The anonymized transcripts are available on request to the corresponding author subject to signature of a data confidentiality agreement.

AUTHOR RESPONSE

Data cannot be shared publicly. In this qualitative study, the transcripts are the data. The transcripts, even if anonymized, risk the confidentiality of the person who provided the information / perspective. In the patient information sheet and consent form that was signed, we do not have the consent to share the transcript from the interviews. Hence, sharing the transcript will breach the compliance with the protocol approved by the ICMR-NIE’s institute human ethics committee. Data are available from the ICMR-NIE’s institute human ethics committee (contact via corresponding author) for researchers who meet the criteria for access to confidential data. The anonymized transcripts are available on request to the corresponding author subject to signature of a data confidentiality agreement.

The same has been reflected in the data availability statement in the tail section of the manuscript. 

Journal Requirements:

ICMR-NIE led USAID/JSI supported TB ACF evaluation project is a collaborative effort involving ICMR-National Institute of Epidemiology (ICMR-NIE), Chennai, India (lead); USAID India, New Delhi, India; JSI India, New Delhi, India; The WHO Country Office for India, New Delhi, India; and Central TB Division, Ministry of Health and Family Welfare, Government of India, New Delhi, India. The authors sincerely thank the support and contribution received from all the state TB cells, district TB cells and the WHO NTEP medical consultant network in India. We also acknowledge the contribution by the interns of ICMR NIE, Chennai, India who provided data transcription support: Mr SM Aakash, Ms M Nisha, Ms Archita Govardhana, Ms Suhana Khatoon B, Ms Aishwarya Dhumale and Mr Mahesh Gomasa. 

This study was commissioned by India’s national TB elimination program, with funding support from USAID through John Snow International (JSI) under the TB Implementation Framework Agreement (TIFA). The contents of this study document are the authors' sole responsibility and do not necessarily reflect the views of USAID or the United States Government.

AUTHOR RESPONSE

We have removed funding information from manuscript file and ensured correct statement in the manuscript submission website and mentioned the same in the cover letter. We have removed funding related information any other parts of the manuscript. 

The correct updated funding statement is as follows

ICMR-NIE led USAID/JSI supported TB ACF evaluation project is a collaborative effort involving ICMR-National Institute of Epidemiology (ICMR-NIE), Chennai, India (lead); USAID India, New Delhi, India; JSI India, New Delhi, India; The WHO Country Office for India, New Delhi, India; and Central TB Division, Ministry of Health and Family Welfare, Government of India, New Delhi, India. This study was commissioned by India’s national TB elimination program, with funding support from USAID through John Snow International (JSI) under the TB Implementation Framework Agreement (TIFA). There was no additional external funding received for this study. The contents of this study document are the authors' sole responsibility and do not necessarily reflect the views of USAID or the United States Government.

Reviewer #1 (Please note: The line numbers mentioned in our response are the line numbers from the “Revised Manuscript with Track Changes”)

REVIEWER

The manuscript, “Bridging the “know-do” gap to improve active case finding for tuberculosis in India: A qualitative exploration into program staffs’ perspectives” describes the enablers, barriers, and suggested solutions to improve TB active case finding India from NTEP staff (provider) perspective.

The authors have described well the introduction, and results. The methodology and the discussion need some improvement. The findings have public health importance to understand what factors may influence TB active case findings. The findings may also have importance in the context of infection prevention and control in community settings. However, the manuscript would require substantial revision to make it clearer and reader friendly.

The manuscript may be appropriate for this journal and could be considered for publication in PLOS ONE after addressing following issues.

AUTHORS’ RESPONSE

Thank you very much for the comment. We appreciate your constructive comments.

REVIEWER 

Abstract:

The objective could be merged with the background section for clarity and conciseness 

AUTHORS’ RESPONSE

Thank you very much for the suggestion. The objective has been merged with the background as suggested. (line 25-32 of revised manuscript with track changes)

REVIEWER 

Abstract Methods:

Additional details are required in this section. The analysis section is very vague and warrants more information. This includes clarification on whether the interviews were recorded, how key informants were selected and why, the type of questions asked, who transcribed the recordings, whether the transcriptions were verbatim, how themes were identified, and if there were any pre-defined themes. Additionally, the resolution of inconsistencies between the two individuals involved in thematic analysis should be explained.

AUTHORS’ RESPONSE

Thank you very much for the comment. These mentioned additional details have been incorporated into the methods section of the abstract. (line 35-44 of revised manuscript with track changes)

REVIEWER 

Abstract Results:

Clarify the term "Ni-kshay" as an international audience may not understand it. The conclusion regarding revising ACF guidance is too generic. Authors should specify the changes required based on the study findings

AUTHORS’ RESPONSE

Thank you very much for the comment. The term “Ni-kshay” has been clarified in the result section of the abstract. (line 55-56 of revised manuscript with track changes)

Additionally, we have clarified the revision required in ACF guidance in the conclusion section of the abstract. (line 60-65 of revised manuscript with track changes)

REVIEWER 

Main manuscript-

Introduction- The statement “Based on the data available, we used three derived ACF quality indicators in our analysis (see Table 1) [16]. We observed that comprehensive mapping of the high-risk populations in the districts was not done at the beginning of the year.” sounds more like a description of the study methods and should be revised accordingly.

AUTHORS’ RESPONSE

Thank you for your comment. 

We would like to clarify that this national-level evaluation of TB-ACF was conducted in a phase-wise manner. 

Under the first phase (secondary data analysis) of this evaluation, we assessed the frequency, scale and quality of ACF at the district, state and national level for the year 2021 using routinely collected secondary ACF data in Ni-kshay (TB information management system under NTEP). The results have been published in the Global Health Action journal and the reference has been provided. 

Due to data related issues, we were not able to calculate all the ACF quality indicators. In the main text, we have clarified how we generated the revised indicators (using the data available) and interpreted the results of the first phase, based on which broad open-ended questions were developed for this descriptive qualitative study. Hence, we find it apt to provide clarification in the introduction section of the main text. 

The study method used for descriptive qualitative study has been provided in the methods section.

REVIEWER

Study settings- Estimations regarding the number of sub-district (block) level administrative TB units and peripheral health institutions (public and private) providing TB diagnosis and treatment services should be provided. The study location, including the names of the districts and reasons for their selection, should be mentioned. Avoid unnecessary information.

AUTHORS’ RESPONSE

Thank you for your comment. 

We have now provided the number of sub-district (block) level administrative TB units and peripheral health institutions (public and private) providing TB diagnosis and treatment services. (line 143-48 of revised manuscript with track changes) 

The names of districts and reason for their selection (purposive) has been mentioned under the sub-section “study population”. 

Other details in the study setting in the current form are required for an international audience to understand what happens under ACF routinely within the NTEP. We have also provided information on TB ACF guidelines released by the Central TB Division in India. We also refer to these points clarified in the settings, later in the methods, results and discussion. Hence, we find this is apt to be included in the study setting.

Considering the flow of the write up, the author group has decided to retain sampling strategy, purposive selection of districts along with reason for selection in the study population sub-section. Reason has been provided below. Study population includes description by time place and person. Hence, the place is part of the study population (the district/state to which the study participants belong). Additionally, we followed purposive sampling strategy to include these districts. The purpose for including these districts has also been mentioned in the study population sub-section. As sampling is part of study population subsection, we decided to retain this aspect (sampling strategy and district included) under study population. We hope this is fine. 

REVIEWER Study population- Focus solely on describing the study population in this section.

“These 30 NTEP districts from nine states formed a convenient, accessible sample for the qualitative systematic enquiry (second phase of the project). Among these 30, six districts (from six different states) were purposively chosen: Ahmedabad Municipal Corporation (Gujarat), Jaipur I (Rajasthan), Aurangabad (Bihar), East Khasi Hills (Meghalaya), Deogarh 167 (Odisha), Pudukkottai (Tamil Nadu) (see Fig 2)” these should go under the study settings.

AUTHORS’ RESPONSE

Thank you for your suggestion. This is a good idea which we considered. However considering the flow of the write up, the author group has decided to retain this in the study population sub-section. Study population includes description by time place and person. Hence, the place is par

---

## [Decision Letter · Decision Letter 1]

28 Jun 2024

PONE-D-23-43822R1Bridging the “know-do” gap to improve active case finding for tuberculosis in India: A qualitative exploration into national tuberculosis elimination program staffs’ perspectivesPLOS ONE

Dear Dr. Shewade,

Thank you for submitting your manuscript to PLOS ONE. After careful consideration, we feel that it has merit but does not fully meet PLOS ONE’s publication criteria as it currently stands. Therefore, we invite you to submit a revised version of the manuscript that addresses the points raised during the review process.

Editor:

1. Clarify the term "Ni-kshay" in the abstract as an international audience may not understand it

2. Specify the changes required in ACF guidance based on the study findings in the abstract conclusion

3. Revise the statement about using derived ACF quality indicators in the introduction, as it sounds more like a description of the study methods

4. Consider quantifying some findings from the 34 interviews to help understand the relevance and importance of issues

5. Organize the analysis better, as numerous themes are clubbed into a few categories making it hard to follow

6. Highlight in the discussion that ACF strategies need to be implemented based on the local context

7. Provide details on how consent was obtained for the second round of interviews and whether it was planned in the protocol

8. Explain the steps taken to maintain confidentiality and mention any refusals for the second round of interviews

9. State the number of repeated interviews conducted in the results

10. Provide the average length of interviews in the results section

Reviwer 2:

Authors have addressed all the comments, however the manuscript requires considerable editing to get in a publishable form and see the flow. please see the attachement

We look forward to receiving your revised manuscript.

Kind regards,

Alireza Goli

Academic Editor

PLOS ONE

Journal Requirements:

Additional Editor Comments:

Editor:

1. Clarify the term "Ni-kshay" in the abstract as an international audience may not understand it

2. Specify the changes required in ACF guidance based on the study findings in the abstract conclusion

3. Revise the statement about using derived ACF quality indicators in the introduction, as it sounds more like a description of the study methods

4. Consider quantifying some findings from the 34 interviews to help understand the relevance and importance of issues

5. Organize the analysis better, as numerous themes are clubbed into a few categories making it hard to follow

6. Highlight in the discussion that ACF strategies need to be implemented based on the local context

7. Provide details on how consent was obtained for the second round of interviews and whether it was planned in the protocol

8. Explain the steps taken to maintain confidentiality and mention any refusals for the second round of interviews

9. State the number of repeated interviews conducted in the results

10. Provide the average length of interviews in the results section

Reviwer 2:

Authors have addressed all the comments, however the manuscript requires considerable editing to get in a publishable form and see the flow. please see the attachement

Reviewers' comments:

Reviewer's Responses to Questions

**Comments to the Author**

1. If the authors have adequately addressed your comments raised in a previous round of review and you feel that this manuscript is now acceptable for publication, you may indicate that here to bypass the “Comments to the Author” section, enter your conflict of interest statement in the “Confidential to Editor” section, and submit your "Accept" recommendation.

Reviewer #2: All comments have been addressed

2. Is the manuscript technically sound, and do the data support the conclusions?

Reviewer #2: Yes

3. Has the statistical analysis been performed appropriately and rigorously? 

Reviewer #2: N/A

4. Have the authors made all data underlying the findings in their manuscript fully available?

Reviewer #2: Yes

5. Is the manuscript presented in an intelligible fashion and written in standard English?

Reviewer #2: Yes

6. Review Comments to the Author

Reviewer #2: Authors tried to address all the comments that I raised. However, further editing is required to make the manuscript publishable. I have no further technical comments.

7. PLOS authors have the option to publish the peer review history of their article (what does this mean?). If published, this will include your full peer review and any attached files.

Reviewer #2: **Yes: **Sachin Atre

---

## [Author Response · Author response to Decision Letter 1]

9 Jul 2024

Editor Comments

GENERAL RESPONSE BY AUTHORS

The indicators in table 1 and Fig 1 include both indicators for ACF scale as well as ACF quality. We have used this opportunity to clarify this in the narrative of introduction and in title / body of Figure 1 and Table 1. Table 1 refers to the derived indicators used in our previously published secondary data analysis of ACF (year 2021) – this was because we did not have all the aggregate numbers (as depicted in Fig 1) to calculate all the indicators. 

ACF scale means, within an ACF cycle, what proportion of mapped high-risk population (it should be adequately mapped) was screened. Of those screened, how many were diagnosed (NNS) is the indicator for ACF quality. If NNS is high, then one may look at the specific ACF quality indicators like – proportion of presumptive among screened, proportion of tested among presumptive and proportion diagnosed among tested. Please refer to Fig 1 for all the indicator of ACF scale and quality. 

We have also used this opportunity to correct some spelling mistakes here and there. 

Please note that Figure 3 has been revised based on reviewer comments.

COMMENT

1. Clarify the term "Ni-kshay" in the abstract as an international audience may not understand it

RESPONSE

We have clarified on first use in both abstract and narrative text. Please see lines 44 and 93 of revised manuscript with track changes. Ni-kshay is the electronic TB information management system under NTEP

COMMENT

2. Specify the changes required in ACF guidance based on the study findings in the abstract conclusion

RESPONSE

In abstract conclusion, we have already specified the changes required in ACF guidance which are as follows: emphasize capacity building, need to carry out ACF in high-risk (not general) population, quality control-linked incentives, and regular implementation monitoring of the activities.

COMMENT

3. Revise the statement about using derived ACF quality indicators in the introduction, as it sounds more like a description of the study methods

RESPONSE

This study plans to go in depth into the findings of our national level assessment of quality of ACF using aggregate secondary data for the year 2021. The section the editor is referring to refer to this previous study that has been cited as reference 16. The derived ACF quality indicators are also with reference to the reference 16.

However, to address reviewer comment we have revised the statement (changed from active to passive voice). We hope it is now fine. 

COMMENT

4. Consider quantifying some findings from the 34 interviews to help understand the relevance and importance of issues

RESPONSE

We thank the editor for the comment. This being a qualitative study, the aim is to not quantify but to explore in detail the reasons for suboptimal quality (from provider perspective) and suggested solutions to improve (from provider perspective). As shared in lines 249-71 of revised manuscript with track changes (subsection data management and analysis), we carried out manual descriptive thematic analysis. However content analysis, on the other hand, involves quantifying data by counting the frequency of specific terms or concepts and organizing them. Our aim was not to quantify but explore the reasons for suboptimal ACF quality that was summarized as themes. We sincerely hope this is fine. 

COMMENT

5. Organize the analysis better, as numerous themes are clubbed into a few categories making it hard to follow

RESPONSE

We thank the editor for the comment. Among enablers, we have six themes under two categories. We have now classified them under three categories. During a review based on your comment, we felt it would be right to divide the category ‘operational planning and stakeholder engagement’ into two separate categories. We have made edits to reflect this in revised figure 3 and the narrative text (see lines 297-363 of revised manuscript with track changes). 

We reviewed the categories and themes under perceived barriers and think they are fine in the current form. 

We reviewed the categories and themes under suggested solutions (as perceived by NTEP staff) and made some minor edits (lines 470-540 of revised manuscript with track changes) in the classification of themes under categories. 

We have made edits here and there in the results narrative to improve clarity.

COMMENT

6. Highlight in the discussion that ACF strategies need to be implemented based on the local context

RESPONSE

We thank the editor for the comment. We have clarified this in lines 618-19 and 668-70 of revised manuscript with track changes (discussion section).

COMMENT

7. Provide details on how consent was obtained for the second round of interviews and whether it was planned in the protocol

RESPONSE

Yes. Repeat interviews (second round of interviews) were planned in the protocol. If required, we had proposed to conduct repeat interviews (second round of interviews) in the protocol. The patient information sheet that was provided during written informed consent (during first interview) mentioned that the participants may be contacted for repeat interviews. Hence, there was no need for consent for repeat interviews.

COMMENT

8. Explain the steps taken to maintain confidentiality and mention any refusals for the second round of interviews

RESPONSE

We have explained the steps in maintain confidentiality in lines 133-35 of revised manuscript with track changes (under ethics). Additionally steps were taken to ensure confidentiality while reporting the results as well. 

There were no refusals to participate (initial or repeat interviews). This has been clarified in lines 211 and 235 of revised manuscript with track changes. 

COMMENT

9. State the number of repeated interviews conducted in the results

RESPONSE

We have included this information in the first line of results narrative and in the footnotes of table 2. Please refer to revised manuscript with track changes.

COMMENT

10. Provide the average length of interviews in the results section

RESPONSE

We have included this information in the first para of results narrative and in table 2. Please refer to revised manuscript with track changes.

Reviewer 2:

GENERAL RESPONSE BY AUTHORS

The indicators in table 1 and Fig 1 include both indicators for ACF scale as well as ACF quality. We have used this opportunity to clarify this in the narrative of introduction and in title / body of Figure 1 and Table 1. 

ACF scale means, within an ACF cycle, what proportion of mapped high risk population (it should be adequately mapped) was screened. Of those screened, how many were diagnosed (NNS) is the indicator for ACF quality. If NNS is high, then one may look at the specific ACF quality indicators like – proportion of presumptive among screened, proportion of tested among presumptive and proportion diagnosed among tested. 

We have also used this opportunity to correct some spelling mistakes here and there. 

Please note that Figure 3 has been revised based on reviewer comments.

COMMENT

Authors have addressed all the comments, however the manuscript requires considerable editing to get in a publishable form and see the flow. please see the attachement

RESPONSE

We thank the reviewer for the comment. Regarding your comments in the attachment, we have addressed them, point by point, below

COMMENT

The study is from a commissioned TB ACF evaluation project by India’s National TB Elimination (NTEP). India is trying to explore implementation of ACF, however, considering the costs and comparative insignificant case finding, passive case finding remained the cornerstone of NTEP for decades. The present study brings out some experiences based on qualitative research which is an important aspect. It is also important that authors addressed “know-do gap” in the ACF implementation. However, in my opinion in the current form it has several limitations in terms of generalizability and overall analysis. Authors may need to undertake a detailed revision 

RESPONSE

We have noted this. We have addressed your comments, point by point, below

COMMENT

The title should reflect “NTEP” program otherwise just stating program does not give an idea to readers.

RESPONSE

We thank the reviewer for the comment. NTEP stands for national tuberculosis elimination program. The title already has the phrase ‘national tuberculosis elimination program’.

We had received this comment during round one of review and in line with this we had changed the title during the first round of review. We are intrigued. Has the reviewer viewed the original submission files (did not have national tuberculosis elimination program in the title) and not the revised version (round one) that had national tuberculosis elimination program in the title?

COMMENT

Abstract: Objective should not include suggested solutions

RESPONSE

We thank the reviewer for the comment. However, we beg to differ here. The objectives of this study were to explore the reasons for suboptimal ACF quality and suggested solutions, both from NTEP staff perspective. Hence, we have retained this as the objective. While we have analyzed and presented the findings in results (that includes suggested solutions from NTEP staff-provider perspective), we have also provided our recommendations in the discussion (see lines 618-62 of revised manuscript with track changes).

COMMENT

34 key informant interviews should go as a part of methods. 

RESPONSE

We thank the reviewer for the comment. However, we beg to differ here. The number of interviews are guided by saturation and at the time of implementing the study or planning this study (what we did? – Methods), we were not sure of the exact numbers. Hence, this should go in the results section. This is more of a style issue. I have seen many authors mention the exact number of interviews in the methods itself. However, the author group, for reasons mentioned above, have provided the exact number of interviews along with repeat of interviews in the results section. We hope this is fine. 

COMMENT

Authors mentioned that they performed manual analysis, but in my opinion, it may sometimes be biased. Why did authors not use qualitative data management programs such as MAXQDA or Nvivo etc.? That would provide more objective analysis.

RESPONSE

We thank the reviewer for the comment. Usually softwares are recommended when one needs to analyse a lot of data. Considering the number of interviews (approx. 30-40), we performed the analysis manually. The number of interviews were not huge enough to mandate the use of qualitative data analysis softwares and we are comfortable with manual analysis. MS Word with comments function on the right, with highlighting was used to identify codes and apt quotes. These were later combined into themes and categories. This was done independently by two authors and then a third author was involved, if required, to review the results and sort out discrepancies in codes and themes. The same has been clarified in lines 249-71 of revised manuscript with track changes (data management and analysis subsection). To be transparent, we have reported all points under data analysis section (following COREQ). 

COMMENT

Lines 78-79: Authors mentioned “ACF did not have better outcome rates as compared to Passive case finding”. This needs to be interpreted carefully. Treatment outcomes do not necessarily always depend upon ACF and early diagnosis, but the entire treatment course as well. So, the comparison needs to be made considering various points in both methods. The rationale for undertaking qualitative research is not sufficiently stated. 

RESPONSE

The line refers to inference drawn from a previous study conducted in a project setting in India (project Axshya) and this is not a finding of this study. The previous study is reference number 11.I am reproducing the para below

“During 2013-15, under The Global Fund-supported project Axshya (Axshya means without TB in Sanskrit) in India, ACF among high-risk populations improved case notification rates (CNRs) at the TB unit level compared to TB units implementing PCF (passive case finding) only [8]. During 2016-17, under the same project, ACF reduced health-system level diagnosis delay at the national level. ACF-detected patients incurred lower costs and lower chances of catastrophic costs during TB diagnosis than PCF-detected patients. ACF-detected patients did not have significantly better treatment outcomes when compared to PCF-detected patients [9–11]. One of the limitations was the lack of qualitative systematic enquiry to explore the provider and beneficiary perspectives for the above findings [12].”

COMMENT

The real problem with ACF implementation is whether incentives are going to continue as that would decide the sustainability of any program. 

RESPONSE

We agree with the comment. Incentives should continue but they should be quality control linked. We have clarified this in detail in the discussion section (see lines 623-37 of revised manuscript with track changes) and have a separate figure for this as well as to what steps to follow to ensure quality control linked incentives (also see Fig 5). 

COMMENT

Methods section should be revised thoroughly as how actually they did the data management and analysis. Authors may describe how content analysis underpinned current research. Authors should describe the detailed method of thematic analysis.

RESPONSE

We thank the reviewer for the comment. To be transparent, we have reported all points under data collection and data analysis section (following COREQ). For data collection, please refer to lines 218-47 of revised manuscript and for data management and analysis, please refer to lines 249-71 of revised manuscript with track changes. 

For further clarity, as requested by the reviewer, we have added appropriate references under the data management and analysis section (see ref 24 and 25 in revised manuscript). In programmatic settings where the researchers want to explore the reasons for a certain problem, thematic analysis is the most appropriate. Hence, content analysis is the theoretical underpinning. Thematic analysis is appropriate for determining solutions to real world problems and we have clarified it in lines 249-71 of revised manuscript with track changes. 

There are five theoretical underpinnings: content analysis (to systematically organize data into a structured format)., grounded theory, phenomenology, ethnography, and discourse analysis. Within the theoretical framework of ‘content analysis’, as a methods of data analysis, we decided to perform thematic analysis, not content analysis. Our interest was to explore themes around reasons for poor ACF quality and suggested solutions to improve it, both from NTEP manager and implementers’ perspective. We were not interested in quantifying the reasons, and this was not our aim. 

COMMENT

Authors mentioned in the abstract that two investigators did a manual descriptive thematic analysis of the transcripts, independent of each other. A third investigator reviewed the same. It is unclear whether there were consistency or discrepancies in this method and how third person contributed. Authors may want to throw light on these issues.

RESPONSE

We thank the reviewer for highlighting this point. The third person reviewed the data if there were discrepancies in the codes generated and we then finalized the codes and themes based on mutual discussion and consensus. A third experienced investigator reviewed the analysis to reduce subjective bias and increase interpretive credibility. The third investigator also discussed and resolved any inconsistencies in the codes and themes generated among the two investigators. We have clarified this in lines 249-71 of revised manuscript with track changes.

COMMENT

Lines 218-220: “The data collected from project research assistants was about ACF implementation by NTEP and not about their own performance. Hence, we do not think that this relationship between investigator and project research assistants would have affected the richness of data”. This is not clear. 

RESPONSE

In COREQ guidelines, it is recommended that the relationship between interviewer and interviewe

---

## [Decision Letter · Decision Letter 2]

19 Aug 2024

Bridging the “know-do” gap to improve active case finding for tuberculosis in India: A qualitative exploration into national tuberculosis elimination program staffs’ perspectives

PONE-D-23-43822R2

Dear Dr. Shewade,

We’re pleased to inform you that your manuscript has been judged scientifically suitable for publication and will be formally accepted for publication once it meets all outstanding technical requirements.

Kind regards,

Alireza Goli

Academic Editor

PLOS ONE

Additional Editor Comments (optional):

Reviewer 2:

Authors have satisfactorily addressed all the comments. There are no further comments from my side. But the manuscript needs further editing.

Reviewers' comments:

Reviewer's Responses to Questions

**Comments to the Author**

1. If the authors have adequately addressed your comments raised in a previous round of review and you feel that this manuscript is now acceptable for publication, you may indicate that here to bypass the “Comments to the Author” section, enter your conflict of interest statement in the “Confidential to Editor” section, and submit your "Accept" recommendation.

Reviewer #2: All comments have been addressed

2. Is the manuscript technically sound, and do the data support the conclusions?

Reviewer #2: Yes

3. Has the statistical analysis been performed appropriately and rigorously? 

Reviewer #2: N/A

4. Have the authors made all data underlying the findings in their manuscript fully available?

Reviewer #2: Yes

5. Is the manuscript presented in an intelligible fashion and written in standard English?

Reviewer #2: Yes

6. Review Comments to the Author

Reviewer #2: Authors have satisfactorily addressed all the comments. There are no further comments from my side. But the manuscript needs further editing.

7. PLOS authors have the option to publish the peer review history of their article (what does this mean?). If published, this will include your full peer review and any attached files.

Reviewer #2: **Yes: **Dr. Sachin Atre

---

## [Editor Report · Acceptance letter]

3 Sep 2024

PONE-D-23-43822R2 

PLOS ONE

Dear Dr. Shewade, 

I'm pleased to inform you that your manuscript has been deemed suitable for publication in PLOS ONE. Congratulations! Your manuscript is now being handed over to our production team.

Kind regards, 

on behalf of

Dr. Alireza Goli 

Academic Editor

PLOS ONE